# Physical Activity Energy Expenditure Predicts Quality of Life in Ambulatory School-Age Children with Cerebral Palsy

**DOI:** 10.3390/jcm11123362

**Published:** 2022-06-11

**Authors:** Jinuk Lee, Min-Hwa Suk, Soojin Yoo, Jeong-Yi Kwon

**Affiliations:** 1Department of Physical & Rehabilitation Medicine, Samsung Medical Center, Sungkyunkwan Universitiy School of Medine, Seoul 06351, Korea; jinuk8780.lee@samsung.com; 2Department of Physical Education, Seoul National University of Education, Seoul 06639, Korea; minhwasuk7@gmail.com; 3Department of Health and Human Performance, University of Texas, Rio Grande Valley, Edinburg, TX 78539, USA; soo.yoo@utrgv.edu

**Keywords:** cerebral palsy, quality of life, physical activity, accelerometry

## Abstract

Background: Participation in physical activities is positively associated with better quality of life in children with cerebral palsy (CP). The objective of this study was to elucidate the relationship between the intensity of habitual physical activity (HPA) measured with an accelerometer and health-related quality of life (HRQOL) in school-age children with CP. Method: A secondary analysis of the cross-sectional data of 46 ambulatory children with CP was conducted. The participants wore an accelerometer for seven days to measure HPA: activity counts (counts/min) and physical activity energy expenditure (PAEE, kcal/kg/day), as well as %moderate-to-vigorous intensity physical activity (%MVPA), %light intensity physical activity (%LPA), and %sedentary physical activity (%SPA) were measured. Pediatric Quality of Life Inventory (PedsQL) 4.0 Generic Core Scales and Child Health Questionnaire Parent Form 50 Questions (CHQ-PF50) were used to measure HRQOL. A Pearson analysis and a hierarchical regression analysis were performed. Results: PAEE significantly predicted the results of the PedsQL(child) physical domain (β = 0.579, *p* = 0.030), PedsQL(child) emotional domain (β = 0.570, *p* = 0.037), PedsQL(child) social domain (β = 0.527, *p* = 0.043), and PedsQL(child) total (β = 0.626, *p* = 0.017). However, other HPA parameters could not predict any other HRQOL. Conclusions: PAEE could be used as a biomarker in studies on HRQOL and HPA in ambulatory school-age children with CP.

## 1. Introduction

A previous systematic review reported that participation in physical leisure activities (e.g., biking, horseback riding, and organized activities) was positively associated with an improved quality of life in children with neurodevelopmental disabilities [1]. Children with cerebral palsy (CP) who engage in physical leisure activities are expected to have a higher quality of life (QOL). Furthermore, it has been reported that exercise training can improve health-related QOL (HRQOL) in children and adolescents with CP [2].

Currently, digital devices are widely used to monitor the amount and intensity of physical activity. Recently, accelerometers have been introduced and used to measure habitual physical activity (HPA) in children with CP [3,4]. These devices record and analyze the extent of physical activity undertaken by the children and classify physical activities as vigorous, moderate, light, and sedentary according to their activity level. In addition, the software of these devices has a conversion formula that allows for the estimation of calories burned during physical activity.

There have been several reports indicating a relationship between motor capacity and HPA in children with CP. Recently, Suk et al. [5] reported that motor capacity showed a mild-to-moderate correlation with the HPA in school-aged children with CP. However, few clinical studies have explored the relationship between HPA and HRQOL. Keawutan et al. [6] reported that HPA was not associated with parent-reported QOL in five-year-old children with CP. However, in another study, HPA was correlated with better HRQOL scores, especially in the physical and social domains [7]. The authors of these studies believe that these two clinical outcomes, HRQOL and HPA, are closely linked. Therefore, we hypothesized that there would be a relationship between these two outcomes in school-aged children with CP who are capable of reporting HRQOL by themselves. Therefore, this study aims to elucidate the relationship between HPA and HRQOL in ambulatory school-aged children with CP. Thereafter, we endeavor to find the HPA parameters that were most representative of HRQOL in children with CP.

## 2. Materials and Methods

### 2.1. Study Design

This study is a sub-study of the clinical trial on the effects of hippotherapy on physical fitness and attention in CP (ClinicalTrial.gov NCT03870893) performed between August 2017 and December 2019 at Samsung Medical Center in Seoul, Korea. A secondary analysis of cross-sectional data from ambulatory school-age children with CP (*n* = 46) was conducted. Written informed consent was obtained from the participants, and their parents or guardians before screening. The Institutional Review Board of the Samsung Medical Center (registration number: 2017-06-045) approved this study.

### 2.2. Participants

Forty-seven children with CP participated in the study. As one child refused to undergo screening, data from 46 children were included in further analysis. The baseline characteristics of the participants are presented in Table 1. There were 22 girls and 24 boys with a mean age 7.48 years, SD 1.56 years. The Gross Motor Function Classification System (GMFCS, level I, II, or III) was used to categorize their level of physical ability. The inclusion criteria were as follows: (1) diagnosis of CP; (2) GMFCS level I, II, or III; (3) age between 6 and 12 years; and (4) body weight <35 kg. The exclusion criteria were (1) selective dorsal rhizotomy or orthopedic surgery in the last 1 year, (2) injection of botulinum toxin in the last 3 months, (3) poor visual acuity, (4) uncontrolled seizures, (5) hearing impairment, (6) severe intellectual disability, (7) hip dislocation, (8) Cobb angle >30° in scoliosis, and (9) unhealed fracture. 

### 2.3. Habitual Physical Activity

HPA was measured using an accelerometer (ActiGraph, Pensacola, FL, USA). The ActiGraph model used in this study is a GT3X triaxial accelerometer monitor set at 5 s epochs. The GT3X accelerometer is reliable and valid for monitoring physical activity in children with CP and is appropriate for rehabilitation research [8]. The participants were instructed to wear the accelerometer around the waist with a belt for seven days, including at the time of sleep and rest. The device collected data from 10 a.m. to 8 p.m. When entering water, such as swimming, showering, or bathing, the participants were instructed to remove the accelerometer. Parents were asked to record the activities of their children while wearing the devices so that we could confirm the activity by matching the record from the parent with the data analyzed with the software. The activity count (counts/min) is a mean value based on the vector magnitude recorded by the accelerometer during physical activity, which is a basic parameter indicating physical activity. The larger the count, the higher the intensity level [9]. Physical activity energy expenditure (PAEE) and percentage of time spent in each level of physical activity (%moderate-to-vigorous physical activity, %MVPA; %light physical activity, %LPA; and %sedentary physical activity, %SPA) were obtained from activity counts (counts/min) using ActiLife software (ActiGraph, Pensacola, FL, USA). Furthermore, the PAEE (kcal/kg/day) was calculated using the Freedson model, which is inherently present in the software [10]. Each cut-off value of physical activity intensity was determined using Evenson’s cut-off value model, which demonstrated the highest accuracy in classifying the physical activity level [11]. For this study, 0–100 counts per minute is SPA, 101–2295 counts per minute is LPA, and 2296 or more counts per minute is MVPA.

### 2.4. Health-Related Quality of Life

The QOL parameters used in this study are as follows.

#### 2.4.1. Pediatric Quality of Life Inventory 4.0 (PedsQL 4.0) 

PedsQL is a QOL questionnaire consisting of four domains for subjects aged from 2 to 18 years, comprising a child’s self-report form and a parent-report form. The four domains were physical, emotional, social, and school activity [12]. Both self-report and parent-report forms were scored on a five-point scale from zero (never a problem) to four (almost always a problem). For interpretability, the 0–4 scale items were transformed to 0–100 as follows: 0 = 100, 1 = 75, 2 = 50, 3 = 25, and 4 = 0, which means that higher scores indicate better HRQOL [13]. The PedsQL 4.0 is the only generic pediatric HRQOL measurement instrument for ages 2 to 18; therefore, both self-report and proxy-report versions are present, maintaining the construction of items and scales consistently.

#### 2.4.2. Child Health Questionnaire PF-50 (CHQ-PF50)

The Child Health Questionnaire (CHQ) was created by the Child Health Project in the early 1990s and is a QOL questionnaire targeting children aged 5–18 years [14]. The CHQ is composed of physical and psychosocial domains. In a study using the CHQ PF-50 (*n* = 80, age 5 to 18, CP), it was validated that the cerebral palsy group had a poorer QOL in all domains compared to healthy children [15]. We used the parent-report forms of the CHQ PF-50 (Physical and Psychomotor domains).

### 2.5. Statistical Analysis

SPSS version 27.0 (IBM, Armonk, NY, USA) was used for statistical analysis. Statistical significance was set at alpha = 0.05, power = 80%. Using the Shapiro–Wilk test, we found that the HPA parameters showed a normal distribution. Pearson’s analysis was performed to identify the correlation between HPA and HRQOL. To analyze the effect of HPA (%MVPA, %LPA, %SPA, PAEE, and activity count) on the HRQOL of children with CP, we performed hierarchical regression analysis by controlling the following parameters: CP type (unilateral, bilateral), sex, GMFCS level, and overweight (85th percentile BMI). In the statistical analysis model, SPA and LPA were excluded from the independent variables because of multicollinearity issues.

## 3. Results

The mean and standard deviation of each parameter of HPA were as follows: PAEE (kcal/kg/day) of 4.27 ± 1.36, %SPA 71.94 ± 5.84, %LPA 23.03 ± 4.71, %MVPA 5.03 ± 2.24, and activity counts (counts/min) 878 ± 252.3. In this study, 46 children wore the accelerometer and had no adverse effects. The average accelerometer wearing time was 6.54 days (standard deviation = 0.80), and there was no significant difference between the CP type and GMFCS level. Thirty-two out of the forty-six participants wore the device for seven days, nine wore it for six days, three wore it for five days, and two wore it for four days. All 46 participants wore the device for the minimum required period (over 5 h on four consecutive days). Table 2 shows the Pearson analysis, which measures the correlation between HPA and QOL parameters. The PAEE showed a weak positive correlation with the PedsQL(child) physical domain, PedsQL(child) total, PedsQL(parent) physical domain, PedsQL(parent) total, and CHQ-PF50(physical). %SPA showed a weak negative correlation with CHQ-PF50(physical). %MVPA showed a weak positive correlation with PedsQL(parent) physical domain, PedsQL(parent) social domain, PedsQL(parent) total, and CHQ-PF50(physical). Activity count was positively correlated with the PedsQL(parent) physical domain, PedsQL(parent) total, and CHQ-PF50(physical). According to our hierarchical regression analysis (Table 3), PAEE predicted the PedsQL(child) physical domain (β = 0.579, *p* = 0.030), PedsQL(child) emotional domain (β = 0.570, *p* = 0.037), PedsQL(child) social domain (β = 0.527, *p* = 0.043), and PedsQL(child) total (β = 0.626, *p* = 0.017). However, parent-reported QOL could not be predicted for any of the HPA parameters. (The remainder of the regression analysis results are shown in the Appendix A).

## 4. Discussion

We found a significant relationship between the HPA and HRQOL in school-age children with CP and GMFCS levels I–III. This study is the first to establish that PAEE is a predictive biomarker of HRQOL in school-age children with CP.

Several studies have been conducted on HPA and HRQOL in children and adolescents with CP. Maher et al. [7] reported that HPA, measured using the Physical Activity Questionnaire for Adolescents (PAQ-A), significantly predicted the PedsQL physical domain (β = 6.12, *p* = 0.02) as well as the social domain (β = 9.27, *p* < 0.01) in 70 Australian children and adolescents with CP aged 11–17 years (GMFCS I–V). These findings emphasize the potential benefits of physical activity on the well-being of young people with CP. Bjornson et al. [16] also reported that the physical (β = 0.36, *p* < 0.01), behavioral (β = 0.32, *p* < 0.01), and emotional (β = 0.29, *p* < 0.05) QOL domains of the CHQ-PF87 showed a significant relationship with the Activities Scale for Kids in children aged 10–13 with CP (*n* = 30). These results are similar to ours, showing a positive correlation between the HPA and HRQOL. 

Conversely, Keawutan et al. [6] reported that HPA (measured with accelerometers) was not related to the PedQL(parent) after controlling for motor function in 5-year-old children with CP (*n* = 58, GMFCS I–V). We also found no significant association between any HPA and parent-reported QOL (PedQL(parent) and CHQ-PF50) of children with CP in the hierarchical regression analysis, even though there was a weak positive relationship between HPA and parent-reported QOL in the correlation analysis (see Table 2 and Table 3). We believe that there is a distinct difference between self-reported and parent-reported QOL. Self-reported QOL seems more meaningful in that it better reflects the child’s inner voice. 

Interestingly, we found that PAEE predicted self-reported HRQOL in school-age children with CP. Total energy expenditure is divided into resting energy expenditure and PAEE. There are various methods to measure energy expenditure: calorimetry, doubly labeled water technique, accelerometer, and indirect calculation through oxygen consumption and heart rate monitoring. Of these, the gold standard method is the doubly labeled water technique [17]. In this study, PAEE was calculated from the child’s activity counts reflecting body weight using the Freedson model inherently present in the Actigraph [11]. To achieve the same PAEE, a heavier child must move more and expend more energy in the same amount of time. Suk et al. [5] reported that accelerometer-measured PAEE was significantly reduced in children with GMFCS level III compared with those with GMFCS levels I and II.

To date, there have been few studies in the medical literature on the relationship between PAEE and health-related problems, including HRQOL. Brenner et al. [18] studied the relationship between energy expenditure and QOL in adult hemodialysis patients. The Short Form Health Survey-36 (SF-36) scores were significantly higher in the high-energy expenditure group than in the low-energy expenditure group (SF-36 score of high-energy expenditure group vs. low-energy expenditure group; 58.35 ± 4.49 vs. 42.85 ± 3.86, mean ± standard error, *p* = 0.028). Ekelund et al. [19] investigated the association between PAEE and metabolic syndrome in middle-aged men and women (*n* = 605). The PAEE measurement, calibrated from heart rate monitoring, was negatively associated with progression toward metabolic syndrome (β = −0.0011, *p* = 0.035). In another study, which was multi-ethnicity research conducted in Kenya [20] (*n* = 1084), PAEE was inversely correlated with cardiovascular risk and central adiposity. Adults with CP have been reported to have a higher prevalence of all chronic diseases, such as diabetes, asthma, hypertension, heart disease, and stroke, than adults without CP [21]. Therefore, it is important for individuals with CP to prolong their lifespan and live as healthfully as possible. Cardiorespiratory fitness is a major benefit of a healthy lifestyle because of its inverse relationship with total and cardiovascular mortality [22]. Combining the results of these previous studies with ours, PAEE monitored with an accelerometer has the potential to be used as an important digital biomarker of cardiometabolic syndrome in patients with CP. 

Long-term lifestyle modification is difficult in people with chronic diseases such as CP [23,24]. Medical healthcare, through a digital platform using various wearable devices, is being used for the health surveillance of chronic diseases [25]. Wearable devices that monitor physical activity and/or heart rate are continuously being developed and are expected to be used in the clinical and research fields. The results of this study show that PAEE is a predictor of HRQOL, suggesting that PAEE could be used as an outcome measure in studies on activity participation in children with CP. Furthermore, PAEE can also be used to motivate patients to engage in more activities more effectively by using a device that displays PAEE rather than a device that only indicates activity counts. 

In the previous report, MVPA was positively associated with Gross Motor Function Measure (GMFM)-66 in children at GMFCS II and III (β = 3.237, *p* < 0.01) [5]. Thus, we expected MVPA to predict HRQOL. According to our hierarchical regression analysis, SPA and LPA were excluded because of multicollinearity issues. However, most physical activities in children with CP consist of low-intensity activities (SPA or LPA). Therefore, our study suggests that LPA (and SPA) play crucial roles in the accumulation of overall PAEE in children with CP [26]. Indeed, without reasonable levels of LPA, it is difficult to accumulate high levels of PAEE because children with CP cannot actively perform MVPA due to physical impairment. Therefore, the role of LPA in health should not be underestimated in children with CP. 

This study had several limitations. First, the number of recruits was small, and the geographic characteristics of the participants were specific. As such, it is difficult to generalize the results and future studies are necessary to extend the research to a more diverse population. Second, the number of GMFCS level III children was smaller than that of GMFCS level I and II children (21/19/6, respectively, GMFCS level I/II/III). Third, accelerometers cannot evaluate water activities such as swimming or walking in water. Finally, accelerometer-measured PAEE is likely to underestimate the actual energy expenditure of children with high GMFCS levels (low motor function). When performing the same level of activity, children with high GMFCS levels had higher energy expenditures than those with low GMFCS levels (high motor function) [27]. This reflects lower cardiorespiratory fitness and lower energy efficiency in CP patients with higher GMFCS levels. Since PAEE measured with accelerometers is a derivative of the activity counts, a higher PAEE in children with GMFCS level I simply means that they engaged in more activities and presumably consumed more energy than those with GMFCS levels II or III. Through the development of a wearable digital device that can measure PAEE by reflecting heart rate, it is expected that the PAEE of people with physical disabilities such as cerebral palsy can be measured more accurately.

Since all HPA parameters are involved in calorie consumption and HPA is correlated with QOL, participation in as many activities as possible is recommended. In the future, HPA can be used as a health indicator not only for school-age children with CP but also for all age groups, especially patients with cardiometabolic syndrome. Further research regarding the association between HPA and cardiorespiratory fitness in children with CP is also needed.

Interventions promoting activity and participation are gaining importance [28]. According to the International Classification of Functioning, Disability, and Health, current CP research is increasingly focused on activity and participation rather than the body structure and function [29]. Thus, this study may promote real world activity and participation as well as digital health care surveillance in children with CP.

## 5. Conclusions

The purpose of study was to elucidate the association between the intensity of physical activity and quality of life in school-age ambulatory children with cerebral palsy. Through this study, we found that in ambulatory school-age Asian children with CP (GMFCS I, II, and III), HPA measured with an accelerometer was associated with HRQOL. PAEE predicts the HRQOL of children with CP (age 6 to 12 years, GMFCS I, II, and III), especially children’s perception of their QOL.

## Figures and Tables

**Table 1 jcm-11-03362-t001:** Basal characteristics of the participants.

	Bilateral CP (*n* = 27)	Unilateral CP (*n* = 19)	All (*n* = 46)
Boys (*n*)	15	9	24
Girls (*n*)	12	10	22
Age (years)	7.55 ± 1.80	7.36 ± 1.16	7.48 ± 1.56
Height (cm)	123.0 ± 11.71	124.5 ± 7.55	123.64 ± 10.14
Weight (kg)	25.04 ± 6.19	24.24 ± 4.02	24.71 ± 5.37
BMI (kg/m^2^)	16.37 ± 2.50	15.55 ± 1.55	16.03 ± 2.18
GMFCS level I (*n*)	5	16	21
GMFCS level II (*n*)	16	3	19
GMFCS level III (*n*)	6	0	6

Values are means ± standard deviation. CP: Cerebral Palsy, GMFCS: Gross Motor Function Classification System, BMI: Body Mass Index.

**Table 2 jcm-11-03362-t002:** Pearson analysis of HPA and QOL parameters.

		PedsQL(Child)					PedsQL(Parent)					CHQ-PF50	
		Physical Domain	Emotional Domain	Social Domain	School Activity	Total	Physical Domain	Emotional Domain	Social Domain	School Activity	Total	Physical	Psychosocial
PAEE	*ρ*	0.298	0.285	0.231	0.138	0.302	0.387	−0.072	0.234	0.182	0.345	0.528	0.075
	*p*	0.044 *	0.055	0.123	0.361	0.041 *	0.008 *	0.636	0.117	0.225	0.019 *	0.000 *	0.620
%SPA	*ρ*	−0.162	0.016	0.102	−0.019	−0.024	−0.245	−0.050	−0.235	−0.149	−0.275	−0.306	0.008
	*p*	0.281	0.914	0.500	0.903	0.874	0.100	0.742	0.116	0.324	0.065	0.039 *	0.959
%LPA	*ρ*	0.151	−0.082	−0.155	0.044	−0.010	0.156	0.074	0.146	0.152	0.199	0.168	−0.003
	*p*	0.317	0.590	0.304	0.771	0.948	0.300	0.625	0.333	0.313	0.186	0.264	0.985
%MVPA	*ρ*	0.106	0.129	0.060	−0.044	0.083	0.310	−0.026	0.304	0.067	0.298	0.442	−0.014
	*p*	0.484	0.394	0.694	0.770	0.583	0.036 *	0.866	0.040 *	0.657	0.044 *	0.002 *	0.924
Activity counts	*ρ*	0.176	−0.005	0.006	−0.050	0.048	0.320	0.035	0.245	0.165	0.325	0.492	−0.035
	*p*	0.243	0.976	0.968	0.740	0.750	0.030 *	0.816	0.101	0.273	0.028 *	0.001 *	0.819

PedsQL: Pediatric Quality of Life Inventory, CHQ-PF50: Child Health Questionnaire-Parent Form 50, PAEE: Physical Activity Energy Expenditure, SPA: Sedentary Physical Activity, LPA: Light Physical Activity, MVPA: Moderate to Vigorous Physical Activity. *ρ*: Pearson Correlation Coefficient, * *p* < 0.05.

**Table 3 jcm-11-03362-t003:** Hierarchical regression analysis results.

	Dependent Variables										
	PedsQL(Child) Physical Domain		PedsQL(Child) Emotional Domain		PedsQL(Child) Social Domain		PedsQL(Child) School Domain		PedsQL(Child) Total Score		VIF
	B	β	B	β	B	β	B	β	B	β	
Controlled variables											
Unilateral CP	−2.635	−0.070	−1.752	−0.034	−11.642	−0.212	−15.899	−0.341	−7.284	−0.195	1.943
Girls	9.846	0.266	10.801	0.210	7.962	0.147	9.402	0.204	9.547	0.259	1.163
GMFCS level_II	−5.632	−0.150	−10.494	−0.201	−14.653	−0.266	−19.589	−0.420	−11.684	−0.313	2.276
GMFCS level_III	−17.970	−0.328	−8.315	−0.109	−45.070 *	−0.560 *	−19.467	−0.285	−22.088	−0.404	2.293
Overweight	−11.278	−0.206	1.138	0.015	−9.744	−0.121	8.877	0.130	−3.864	−0.071	1.519
Independent variables											
PAEE	7.917 *	0.579 *	10.871 *	0.570 *	10.588 *	0.527 *	5.122	0.301	8.532 *	0.626 *	3.386
%MVPA	−2.919	−0.352	0.838	0.072	−2.442	−0.201	−1.343	−0.130	−1.656	−0.200	3.588
Activity Counts (counts/min)	−0.010	−0.134	−0.053	−0.517	−0.049	−0.450	−0.019	−0.202	−0.030	−0.403	3.502
R^2^	0.280		0.241		0.309		0.213		0.308		-

VIF: Variance Inflation Factor, CP: Cerebral Palsy, PedsQL: Pediatric Quality of Life inventory, GMFCS: Gross Motor Function Classification System, PAEE: Physical Activity Energy Expenditure, LPA: Light Physical Activity, MVPA: Moderate to Vigorous Physical Activity. * *p* < 0.05; B: unstandardized coefficient; beta: standardized coefficient.

## Data Availability

The data that support the findings of this study are available from the corresponding authors upon request.

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
