# Peer review of "Physical Activity Energy Expenditure Predicts Quality of Life in Ambulatory School-Age Children with Cerebral Palsy"

_jcm, 2022, doi:10.3390/jcm11123362_

Round 1

Reviewer 2 Report

The article presents very precisely defined and strictly defined studies of adolescent patients with cerebral palsy. The results were obtained by conducting studies on a sample of patients identified as school-age children with cerebral palsy.

Generally, the authors put forward a thesis that there are correlations between their physical activity and their mental well-being. In order to proof that in contexts, the work as a whole is well and precisely related to the results of research by other authors.

They use methods of statistical analysis of their results recognised and used by other researchers. The authors use a hierarchical linear model, i.e. an analytical method recommended when there is a high probability of violating the requirement of independence of observation.The requirement of independence is met, with very high probability. On the other hand, one of the criteria for excluding patients from the sample, i.e. severe intellectual disability, may raise doubts. Does this mean that moderate or mild intellectual disability was acceptable? If such patients were allowed in the study, it could have influenced the final results.

The very precise and detailed reference in the Materials and Methods chapter to the research methodology used by the authors and in the Discussion chapter the reference to the obtained results, against the background of others, deserves attention and a positive evaluation.

On the other hand, as critical remark: in Conclusion, the categorical statement that in ambulatory school-age Asian children with CP (GMFCS I, II, and 290 III), HPA measured by an accelerometer was associated with HRQOL. Due to the very limited size of the sample, it would be advisable to indicate the limit of the population from which it was selected and to formulate a more detailed conclusion, not to generalise. After all, the authors write that those with cerebral palsy are affected approximately 2 per 1000 live births. The sample representative of such a large population is much larger. It is advised to put such a limitation as to the obtained results in the conclusion and refer to the mentioned above the criteria for selecting the sample.

Author Response

Response to Reviewer 2 Comments

Point 1:

On the other hand, as critical remark: in Conclusion, the categorical statement that in ambulatory school-age Asian children with CP (GMFCS I, II, and III), HPA measured by an accelerometer was associated with HRQOL. Due to the very limited size of the sample, it would be advisable to indicate the limit of the population from which it was selected and to formulate a more detailed conclusion, not to generalise

Response 1:

We agree that generalizing the results and conclusion is not proper due to the small number of study participants. Therefore, I will emphasize in limitation that the results can properly be applied in only specific population and will add that it is necessary to proceed research on a more diverse population.

chment.